# Krebs Cycle Intermediate-Modified Carbonate Apatite Nanoparticles Drastically Reduce Mouse Tumor Burden and Toxicity by Restricting Broad Tissue Distribution of Anticancer Drugs

**DOI:** 10.3390/cancers12010161

**Published:** 2020-01-09

**Authors:** Sultana Mehbuba Hossain, Syafiq Asnawi Zainal Abidin, Ezharul Hoque Chowdhury

**Affiliations:** 1Jeffrey Cheah School of Medicine and Health Sciences, Monash University Malaysia, Jalan Lagoon Selatan, Bandar Sunway, 47500 Subang Jaya, Selangor, Malaysia; hossainmishu_1990@yahoo.com (S.M.H.); syafiqnawi@gmail.com (S.A.Z.A.); 2Health and Wellbeing Cluster, Global Asia in the 21st Century (GA21) Platform, Jeffrey Cheah School of Medicine and Health Sciences, Monash University Malaysia, Jalan Lagoon Selatan, Bandar Sunway, 47500 Subang Jaya, Selangor, Malaysia

**Keywords:** cyclophosphamide, doxorubicin, carbonate apatite nanoparticles, breast cancer, cytotoxicity, tumor regression, biodistribution, toxicology, blood serum

## Abstract

The morphology, size, and surface area of nanoparticles (NPs), with the existence of functional groups on their surface, contribute to the drug binding affinity, distribution of the payload in different organs, and targeting of a particular tumor for exerting effective antitumor activity in vivo. However, the inherent chemical structure of NPs causing unpredictable biodistribution with a toxic outcome still poses a serious challenge in clinical chemotherapy. In this study, carbonate apatite (CA), citrate-modified CA (CMCA) NPs, and α-ketoglutaric acid-modified CA (α-KAMCA) NPs were employed as carriers of anticancer drugs for antitumor, pharmacokinetic, and toxicological analysis in a murine breast cancer model. The results demonstrated almost five-fold enhanced tumor regression in the cyclophosphamide (CYP)-loaded α-KAMCA NP-treated group compared to the group treated with CYP only. Also, NPs promoted much higher drug accumulation in blood and tumor in comparison with the drug injected without a carrier. In addition, doxorubicin (DOX)-loaded NPs exhibited less accumulation in the heart, indicating less potential myocardial toxicity in mice compared to free DOX. Our findings, thus, conclude that CA, CMCA, and α-KAMCA NPs extended the circulation half-life and enhanced the anticancer effect with reduced toxicity of conventional chemotherapeutics in healthy organs, signifying that they are promising drug delivery devices in breast cancer treatment.

## 1. Introduction

Breast cancer is the most common cancer in women [1] and the second leading cause of global cancer death after lung cancer [2]. Triple-negative breast cancer (TNBC) represents almost 15–20% of breast cancers [3] which can be distinguished by the absence of estrogen receptor (ER), progesterone receptor (PR), and human epidermal growth factor receptor 2 (HER2) on breast cancer cells [1,4].

TNBC is associated with a higher mortality rate due to its heterogeneous and extremely aggressive nature with limited treatment responses, enhanced recurrence, and metastasis, especially in the lungs and brain [1,5,6]. Conventional single mono-chemotherapy is inadequate for treating TNBC, whereas cyclophosphamide (CYP) is highly reported as a neoadjuvant therapy with an anthracycline, microtubule stabilizers, or platinum agents to achieve improved efficacy and maximum benefits [6,7,8]. However, in these therapies, CYP was administered to the patients in a high dose of almost 600 mg/m^2^ [8] to exert its clinical efficacy [9,10], leading to nausea, vomiting, neutropenia [8], and cardiac toxicity [11].

The alkylating agent CYP is a prodrug which is metabolized in the liver by cytochrome P450, producing pharmacologically stable toxic metabolites—acrolein and phosphoramide mustard. These metabolites passively enter into the cell and attach to the guanine base of DNA, which hinders DNA replication by establishing intrastrand and interstrand DNA cross-linking [11,12]. CYP is a widely used chemotherapeutic in cancer treatment due to its direct cytotoxic effect on cancer cells with an immunoregulatory response [13] depending on the dose of the drug, as shown in a mouse model, ranging from 20–200 mg/kg [14,15,16]. 

The main objective of choosing and designing these chemical compounds as chemotherapeutic drugs which are toxic in nature is to destroy all rapidly dividing cells and restrict the spread of cancer. The key shortcoming of this approach is that it not only targets uncontrolled cancer cell proliferation, but also kills the body’s rapidly proliferating normal cells, such as hair follicles and intestinal epithelia, leaving the patient to cope with life-altering side effects [17,18]. This downside of the treatment urged the development of smartly designed nanoparticles (NPs) for targeted delivery of anticancer drugs to the tumor site, while avoiding the normal healthy tissues and organs [19,20]. 

NP-mediated anticancer drug delivery systems currently focus on improving the effectiveness of the chemotherapeutics, by altering the pharmacokinetics of the drugs, delivering neoadjuvant therapies, selectively targeting tumor cells, enhancing the cellular uptake of anticancer agents, releasing the payload inside the tumor microenvironment, and simultaneously ensuring the drug delivery and efficacy [21]. The larger size of NPs prohibits the carrier from crossing the tight junctions of healthy blood vessels; however, the NPs can breach the inflamed or injured tumor site due to leaky junctions, and they can stay in the tumor region due to the absence of lymphatic drainage, allowing the NPs to deliver the anticancer drug in the cancer cell for a prolonged period of time. This phenomenon is known as the enhanced permeability and retention (EPR) effect that could be harnessed for passive targeting [22].

Among the wide variety of NPs, carbonate apatite (CA) NPs and surface-modified CA NPs attracted increasing attention due to their auspicious therapeutic outcomes in nucleic acid and anticancer drug delivery by virtue of a high drug/nucleic acid encapsulation capacity, favorable pharmacokinetics, enhanced cellular uptake, pH-sensitive payload release, efficient transgene expression or gene silencing, and eventually a significant tumor-inhibitory effect both in vitro [23,24,25,26,27,28] and in vivo [29,30,31,32,33,34,35].

CA NPs possess PO_4_^3−^/HCO_3_^−^- and Ca^2+^-rich domains on their outer surface. Due to the heterogeneous surface charge, these NPs provide an ample opportunity to bind drugs via ionic interactions. The drug-loaded CA NPs cross the cell membrane from systemic circulation through an energy-dependent process termed as endocytosis. After the internalization of the cargo, it encounters a highly acidic environment inside the endosomal compartment where the phosphate and carbonate ions in the CA structure tend to consume excess protons from the microenvironment, leading to low pH-dependent dissolution of NPs and consequent drug release [36,37,38]. 

Previously, we successfully synthesized and characterized CA NPs, citrate-modified CA (CMCA) NPs [28,39], succinate-modified CA (SMCA) NPs, and α-ketoglutaric acid-modified CA (α-KAMCA) NPs [40], and we evaluated their particle size, drug-loading efficiency, enhancement in cytotoxicity, and cell viability in a human breast cancer cell line (MCF-7 cells) and a murine breast cancer cell line (4T1 cells). Here, we examined CA, CMCA, and α-KAMCA NPs for in vivo tumor regression, biodistribution, plasma concentration, and toxicology study depending on their promising in vitro results, especially through a cytotoxicity assay in the 4T1 cell line. These 4T1 cells are from an extremely tumorigenic and aggressive mouse mammary adenocarcinoma cell line, resembling many characteristics of human TNBC [41], and they represent an in vivo model for human stage IV breast cancer.

Doxorubicin (DOX) is an anthracycline group chemotherapeutic that prevents DNA replication by breaking down DNA chains during DNA supercoiling, and it also produces free radicals that induce DNA and cell membrane damage. However, DOX causes major side effects such as cardiomyopathy, leading to congestive heart failure and death [42,43,44,45]. In our study, we loaded DOX into CA, CMCA, and α-KAMCA NPs owing to its fluorescent properties, prior to tracking accumulation and distribution of the drug-loaded cargos in different tissues, tumor, and plasma by using fluorescence-based techniques. For the tumor regression study, CYP was incorporated into CA, CMCA, and α-KAMCA NPs, and the NPs without drug were also subjected to a toxicology study to analyze any potential side effects caused by the NPs on the healthy tissues.

## 2. Results

### 2.1. Estimation of Drug Binding Affinity to CA, CMCA, and α-KAMCA NPs

The successful delivery of the drug to the site of action depends on the high drug binding affinity with the minimal amount of materials needed for administration. Furthermore, drug-loaded NPs ensure enhanced solubility leading to improved bioavailability, decreased clearance, and increased stability due to their surface morphology, and a larger surface area with small particle size [46]. The α-KAMCA NPs revealed 80.96% encapsulation efficiency for CYP, while CA NPs exhibited 79.26% and CMCA NPs revealed 76.26% binding affinity (Figure 1A). At 26.11 µg/mL (100 µM) concentration, the mass of CYP loaded into CA, CMCA, and α-KAMCA NPs was around 20.69 µg/mL, 20 µg/mL, and 21.14 µg/mL, respectively, suggesting that all three of these different NPs possess significant and similar drug-loading capacity. In addition, at 10 µM concentration (5.77 µg/mL), DOX showed almost 26.25%, 31.38%, and 36.15% binding affinity for CA, CMCA, and α-KAMCA NPs, respectively (Figure 1B).

### 2.2. In Vivo Anticancer Efficacy

Conventional anticancer drugs are administered frequently in a high dose to maintain an adequate amount of drug available in the tumor site as the drugs are not target-specific, triggering adverse side effects due to toxicity to the healthy tissues. It is, therefore, desirable for NPs to deploy the cargo predominantly to the tumor site to maximize the therapeutic benefits and eliminate it from the body within a certain period of time [47]. To reveal the potential antitumor efficacy of CYP-loaded NPs in breast cancer therapy, a murine breast cancer model was established (Figure 2A) and treated with empty CA NPs, CMCA NPs, and α-KAMCA NPs, as well as various CYP-loaded NPs and a free CYP solution (0.3 mg/kg body weight (BW)) (Figure 2B). While none of the particles with or without drugs interfered with gradual body weight gain, free CYP significantly slowed the gain in comparison to the untreated group, indicating that free CYP, even at a low dose, was highly toxic to the animals, while neither empty NPs nor CYP-loaded NPs possessed any visible toxicity.

Drug-free CA NPs, CMCA NPs, and α-KAMCA NPs were injected intravenously into the tumor-bearing mice to investigate any potential toxic effect of the NPs on the tumor site in comparison to the untreated mice. After 27 days, the tumor size was 2192.20 ± 280.66 mm^3^, 1869.48 ± 212.29 mm^3^, 2115.06 ± 344.67 mm^3^, and 2039.55 ± 232.52 mm^3^ in the untreated, CA NP, CMCA NP, and α-KAMCA NP groups, respectively, indicating that drug-free NPs apparently possess no toxicity to the tumor cells (Figure 3, Figure 4 and Figure 5). 

As shown in Figure 3, the tumor size of the free CYP-treated group continued to grow, albeit at a slower pace compared to the untreated group, indicating limited tumor cell proliferation due to the apoptotic effect of the anticancer drug. The treatment was started from day 14 after tumor inoculation. The tumor size was 145.89 ± 33.53 mm^3^ at day 14, while the tumor size was recorded at 633.33 ± 121.16 mm^3^ for the free CYP-treated mice group at day 27, which was almost 3.5 times less than the untreated group.

The CYP-loaded CA NP- and CYP-loaded CMCA NP-treated groups showed a nearly consistent reduction in tumor volume during the treatment period (Figure 3 and Figure 4). At day 27, the tumor size was 320.96 ± 62.09 mm^3^ (*p* < 0.0001) and 397.51 ± 90.23 mm^3^ (*p* < 0.005) for the CYP-loaded CA NPs and CYP-loaded CMCA NPs, respectively, which was almost two times less than the group treated with only CYP and six times less than the untreated group. 

The CYP-loaded α-KAMCA groups revealed a very interesting result in terms of tumor size reduction (Figure 5). When the first treatment was administered, the tumor volume was 114.47 ± 26.68 mm^3^. After the third day of the first dose, the tumor size was recorded at 71.03 ± 8.44 mm^3^ (*p* < 0.0001), signifying substantial tumor cell apoptosis after the intervention. However, at the end, the tumor size was measured at 127.24 ± 92.51 mm^3^ (*p* < 0.0001), leading to a tumor volume almost five times less than that of the free drug-treated group and demonstrating the best antitumor capacity compared to the other treatment groups.

### 2.3. Biodistribution Studies

NPs as an anticancer drug delivery system offer many advantages; however, a number of issues, such as stability of the drug-loaded NPs in the blood circulation, accumulation in the cancer site, uptake by the cancerous cells, and clearance from the body [48], need to be addressed. In order to determine the distribution patterns of DOX, DOX-loaded CA, DOX-loaded CMCA, and DOX-loaded α-KAMCA NPs, 2-h and 24-h time points were chosen to collect major organs, tumor tissue, and blood from treated mice (Figure 6, Figure 7 and Figure 8). The tumor tissue, organs, and blood from untreated mice were used as a control (Appendix A). 

*Blood Serum:* At 2 h post intravenous (IV) injection (Table 1), the concentration of DOX in the blood serum was significantly higher for CA (538.13 ± 37.01 ng/100 μL), CMCA (339.01 ± 19.64 ng/100 μL), and α-KAMCA (877.21 ± 9.33 ng/100 μL) NPs compared to the free drug. In contrast (Figure 6), the DOX-loaded α-KAMCA NPs showed almost nine-fold (8958.7 ± 340.06 relative fluorescence units (RFU)/100 µL, *p* < 0.0001) higher blood serum concentrations, while CA NPs and CMCA NPs exhibited almost six-fold (5997.1 ± 114.58 RFU/100 µL, *p* < 0.0001) and three-fold (3338.1 ± 114.58 RFU/100 µL, *p* < 0.0001) higher concentrations, respectively, than the free DOX (1008.3 ± 431.41 RFU/100 µL), indicating the role of NPs, particularly the surface-modified ones (α-KAMCA and CMCA NPs), in prolonging the serum half-life of the drug, seemingly by preventing opsonin-mediated reticuloendothelial uptake of NPs, as well as passive tissue distribution and renal clearance of free drugs. The blood serum concentration at 24 h of treatment, 440.28 ± 7.78 ng/100 μL (3029.7 ± 283.62 RFU/100 µL, *p* < 0.0001), 324.75 ± 19.79 ng/100 μL (2217.9 ± 386.47 RFU/100 µL, *p* < 0.0001), and 237.06 ± 27.02 ng/100 μL (1621.3 ± 265.54 RFU/100 µL, *p* < 0.0001), demonstrated a significant increase in the mean residence time (MRT) of DOX-loaded α-KAMCA, CA, and CMCA NPs, respectively, explaining the prolonged circulation time. 

*Tumor:* At 2 h of treatment (Table 2, Figure 7 and Appendix A), α-KAMCA, CMCA, and CA NPs presented 7.3-fold (2.75 ± 0.31 ng/mg), 5.5-fold (2 ± 0.013 ng/mg), and 4.5-fold (1.64 ± 0.06 ng/mg) more accumulation of DOX compared to the free drug treatment (0.37 ± 0.002 ng/mg), which could be due to the higher amount of drug-loaded NPs in the plasma serum, allowing more drug to accumulate in the tumor site. At 24 h of treatment (Table 3, Figure 8 and Appendix A), we determined five-fold (1.79 ± 0.09 ng/mg), 3.6-fold (1.34 ± 0.1 ng/mg), and 2.5-fold (1.04 ± 0.06 ng/mg) more presence of DOX-loaded α-KAMCA NPs, DOX-loaded CMCA NPs, and DOX-loaded CA NPs, respectively, compared to the free DOX treatment (0.37 ± 0.01 ng/mg). This finding can be further explained by the EPR effect via which drug-loaded NPs entered the tumor site by virtue of the leaky vasculature of the tumor blood vessel and stayed for a prolonged period of time due to the impaired lymphatic drainage, enabling NPs to exert a site-specific anticancer effect.

*Heart:* The three apatite-based NPs significantly lowered the distribution of DOX to the heart compared to free DOX at the 2-h and 24-h time points of study (Table 2 and Table 3, Figure 7 and Figure 8, and Appendix A). After 24 h of DOX administration, the concentrations of the drug in the case of DOX-loaded CMCA and DOX-loaded α-KAMCA NPs were 6.59-fold (0.59 ± 0.08 ng/mg) and 7.67-fold (0.58 ± 0.02 ng/mg) lower in the heart, respectively, compared to the free drug (4.28 ± 0.63 ng/mg). Since the size of a drug molecule itself is much smaller than the drug–particle complex, the free drug could easily enter through the tight junction between the cardiac muscle tissues, exerting a toxic effect on the healthy cells, whereas the drug–particle complex was too big to cross the tight junctions of the healthy tissue, resulting in reduced drug accumulation in the heart. 

*Reticuloendothelial Organs:* Tissues enriched with a reticuloendothelial system (RES) such as the liver, lungs, and spleen, and a non-RES organ (the kidney) were analyzed (Figure 7 and Figure 8) to determine the clearance route of the NP-bound DOX and free drug. The highest concentrations of DOX-loaded NPs were observed in the liver and kidney, at almost 1.61 ng/mg at all points of time (Table 2 and Table 3).

*Brain:* Interestingly, drug-incorporated NPs demonstrated a significant accumulation inside the brain at 2 h of treatment (Table 2 and Figure 7, Appendix A), indicating a possibility of crossing the blood–brain barrier (BBB), as well as the blood–cerebrospinal fluid (CSF) barrier (BCSFB), carrying the payload inside the central nervous system (CNS). The BBB, CSF, and BCSFB restrict the entry of toxic compounds and foreign particles inside the CNS, imposing limitations on therapy in the CNS [49]. These data present a promising opportunity to employ NPs to transport therapeutics for treating disorders in the CNS. 

### 2.4. Toxicology Study

To examine in vivo safety, CA NPs, CMCA NPs and α-KAMCA NPs were investigated for one day to explore acute toxicity and for 27 days to inspect sub-chronic toxicity. The inorganic carbonate apatite-based NPs of similar size were injected without a drug to check for any immune response, inflammatory response, and altered hematological factors in the blood. For the acute toxicity study, the mice treated with α-KAMCA NPs showed a significant difference from the untreated group (control = 9.52 ± 1.68) in urea level (14.98 ± 1.82, *p* < 0.01), which might be due to the major clearance pathway for the former NPs (Table 4). Blood hematology tests revealed no significant differences between untreated mice and NP-treated mice groups, except for a slight difference in Na^+^ level for CMCA NPs (155.8 ± 0.84, *p* < 0.003) and α-KAMCA NPs (160.8 ± 2.95, *p* < 0.001), and in Cl^−^ level for CMCA NPs (119.8 ± 1.30, *p* < 0.01) and α-KAMCA NPs (123.2 ± 2.59, *p* < 0.001). CA NPs exhibited no significant difference in blood biomarker profile excluding phosphate level (4.16 ± 1.1, *p* < 0.003). Despite this difference, the levels fell within the normal range. 

For the sub-chronic toxicity study, the urea, Na^+^, Cl^−^, and phosphate levels of CA, CMCA, and α-KAMCA NPs remained the same as for the untreated group of mice; however, a slight difference was noted for alanine transaminase (ALT) level (25 ± 4.47, *p* < 0.02) for CMCA NPs, amylase (Amy) level (1601.4 ± 160.98, *p* < 0.001) for CA NPs, and Ca^2+^ level (2.3 ± 0.28, *p* < 0.05) for α-KAMCA NPs compared to the untreated group (Table 5). 

### 2.5. Morphology Analysis of CYP- and DOX-Loaded CA, CMCA and α-KAMCA NPs

Field emission (FE)-SEM characterization was performed to analyze the surface property, size, and the binding nature of CYP-loaded CA, α-KAMCA, and CMCA NPs, as well as DOX-loaded CA, α-KAMCA, and CMCA NPs (Figure 9). CYP-loaded CA NPs and DOX-loaded CA NPs formed a huge number of spherical-shaped particles with an average size of 251–314 nm and 349–429 nm, respectively (Figure 9A,B). The sizes of CYP-loaded α-KAMCA NPs and DOX-loaded α-KAMCA NPs were around 213–258 nm and 180–212 nm (Figure 9C,D), respectively, with globular shape, and they possessed a more folded surface than the drug-loaded CA NPs, resulting in a large surface area for an efficient amount of drug binding. Interestingly, DOX-loaded α-KAMCA NPs formed branched nanoclusters with numerous tiny particles of around 15–25 nm. However, the sizes of CYP- and DOX-loaded CMCA NPs were in the range of 401–436 nm with a heterogeneous particle shape and size distribution. The size and morphology of CMCA NPs remained unchanged regardless of the drug molecules bound to the particles (Figure 9E,F). 

### 2.6. Protein Corona Analysis

After IV administration of drug-loaded CA, CMCA, and α-KAMCA NPs, an NP–protein complex was immediately formed, which played a crucial role in the systemic circulation, biodistribution, cellular uptake, toxicity, and bioavailability of the drug. The fabricated nanoparticles in 10% fetal bovine serum (FBS)-supplemented Dulbecco’s modified Eagle medium (DMEM) formed a unique protein corona consisting mainly of transport proteins, such as albumin (ALB) protein, serum albumin, serotransferrin, α-fetoprotein, vitamin D-binding protein, uncharacterized protein, apolipoprotein A-II, hemoglobin fetal subunit beta, globin A1, hemoglobin subunit beta, hemoglobin beta, and globin C1, while lacking common opsonins (Table 6 and Table 7). 

Figure 10 and Table 8, Table 9 and Table 10 show the distribution of proteins in the corona where transport proteins made up the largest segment of the protein corona with 60%, 73.33%, and 68.75% for CA, CMCA, and α-KAMCA NPs, respectively, including albumin proteins, which are the most abundant proteins in plasma (also known as dysopsonins that inhibit opsonization), followed by immunoglobulins and coagulation proteins responsible for the alteration of NP opsonization, allocation, and targeting efficiency [50,51,52]. The constituents of the protein corona, shown in the abovementioned figure and tables, display an abundance of transport proteins, including several albumin proteins, but a lack of immunoglobulins, coagulation factors, globulins, complement proteins, and fibrinogen. CA, CMCA, and α-KAMCA NPs were found to bind acute-phase proteins at almost 13.33%, 13.33%, and 12.5%, respectively, which may contribute to opsonization. This finding was further supported by the distribution of NP-encapsulated DOX in the tissues enriched with RES such as the liver, lungs, and spleen (Table 3). Furthermore, the presence of apolipoprotein in the NPs [53] suggests that drug-loaded NPs could be easily taken up by the brain endothelial cells.

### 2.7. Dissolution Study of CA, CMCA, and α-KAMCA NPs at Different pH

The dissolution study of CA, CMCA, and α-KAMCA NPs showed that the NPs were stable at physiological blood pH (7.4) but dissolved in endosomal acidic pH (6.5–5.0) (Figure 11). CA, CMCA, and α-KAMCA NPs were confirmed to show almost 84.4%, 69.41%, and 58% dissolution in pH 6.5, respectively, and 97.2%, 90.29%, and 97% dissolution in pH 5 within 5 min, respectively. Considering the phenomena, CA, CMCA, and α-KAMCA NPs could release the payload very rapidly in an acidic tumor environment (endosomes) and discharge it into the cytoplasm due to the possession of unique pH-sensitive characteristics.

## 3. Discussion

The average particle size/diameter with its polydispersity index (PDI) is a determinant to control the maximum cellular uptake, internalization, biodistribution, drug release profile, and bioavailability of an encapsulated therapeutic compound [54,55,56]. CA NPs, CMCA NPs, and α-KAMCA NPs prepared using a salt precipitation technique resulted in smaller particle sizes which were, respectively, 428.4 ± 21.70 nm, 163 ± 10.96 nm, and 291.5 ± 10.60 nm [28,39]. Drug-loaded NPs showed a particle size ranging from 180–436 nm (Figure 9). In our previous study [28,39,40], we recorded the surface charges for all NP formulations within the range of −9 to −13 mV with a PDI value of 0.553, 0.787, and 0.322 for CA, CMCA, and α-KAMCA NPs, respectively. The PDI values of the nanocarriers pointed out that α-KAMCA NPs are homogeneous (monodisperse) compared to CA and CMCA NPs. The size and PDI value together play pivotal roles in influencing the endocytosis-dependent cellular uptake, especially for in vivo applications.

In contrast, the drug-loading capacity or binding affinity of the NPs depends on the solubility profile of the drug and also the properties of the carrier molecules which involve molecular weight, chemical structure, drug–salt interaction, and the presence of functional groups [22]. CYP showed higher binding affinity ranging from 76–81% toward CA, CMCA, and α-KAMCA NPs at physiological pH (7.4), indicating that the protonated secondary and tertiary cationic amine groups of the CYP structure might bind to the anionic domains of the nanoparticles via electrostatic interaction. CMCA, SMCA (succinate-modified carbonate apatite), and α-KAMCA NPs were prepared by modifying CA with citrate, containing three carboxyl groups, succinate, containing two carboxyl groups, and ketoglutarate, containing one ketone group and two carboxylic groups, respectively. In our previous experiment [28,39,40], we showed that the presence of carboxylic and ketone groups in the chemical structure has an influential role in determining the particle size and the binding affinity of the drugs toward the NPs. The α-KAMCA NPs exhibited the desired particle size with a large number of resultant particles and a folded larger surface morphology with rapid dissolution at acidic pH in the microenvironment, resulting in more significant cytotoxicity than the CA NPs.

The targeted delivery of the drug–NP complex to the specific tissue or organ could be achieved by modifying the size, surface charge, and the surface morphology of the NPs. However, there would be no meaningful outcome if the drug could not be released from the nanoparticle matrix. The release of drug from the nanoparticle-based formulation depends on many factors including pH, temperature, drug solubility, desorption of the surface-bound or adsorbed drug, drug diffusion through the nanoparticle matrix, nanoparticle matrix swelling and erosion, and the combination of erosion and diffusion processes [57,58,59]. The influence of the pH-sensitive dissolution of CA, CMCA, and α-KAMCA NPs ensured drug release in a acidic microenvironment [28,40] in our former study. An in vitro turbidity test (Figure 11) was conducted by modeling physiological pH (7.4) and endosomal acidic pH (6.5–5.0), and the NPs were subjected to these environments to ensure pH sensitivity. The NPs were stable at physiological blood pH (7.4), but rapidly dissolved in the weakly acidic microenvironment. Moreover, all of the NPs released their payload within 5 min at late endosomal pH (pH 5), suggesting possible discharge of the drug into the cytoplasm. The phosphate (PO_4_^3−^) and carbonate ions (CO_3_^2−^) present in the apatite structure could readily accept excess H^+^ ions from the endosomal acidic microenvironment, causing the particles to be dissolved [38]. Furthermore, the cation (Ca^2+^) and anions (PO_4_^3−^, CO_3_^2−^) released from the particles might create osmotic pressure across the endosomal membrane, which in turn could lead to endosome swelling and rupture, releasing the payload into the cytoplasm [23,60]. However, after the dissolution of the NPs in the endosome, the drug might also enter into the cytoplasm via passive diffusion through the endosomal membrane.

Intravenous administration of NPs is the most common route as it can bypass the epithelium absorption barrier by directly entering into systemic circulation [61]. Nanoparticle–protein interaction takes place in circulation prior to distribution into various organs [48,62]. As a result, to deliver the drug, NPs require a perfect targeting device to ensure selective accumulation of the drug in the targeted site while maintaining a low concentration in the healthy organs. The size, surface charge, and morphology of the NPs play an important role in targeting and clearance from the body through the mononuclear phagocytic system (MPS), renal system, and the immune system; however, it is important for the NPs to be eliminated from the body within a predetermined time frame to avoid possible systemic toxicity [48,63,64].

The experimental results reveal that the biodistribution of DOX is greatly altered when delivered through NPs. The NPs significantly enhanced the circulation half-life of DOX in the blood (Figure 6, Appendix A). Among all types of NPs, at 2 h of treatment, DOX-loaded CMCA NPs showed less liver uptake, which could be due to their rapid renal clearance due to their smaller particle size [65], ranging from 36 nm to 72 nm (Figure 7, Appendix A). The α-KAMCA NPs resulted in higher DOX concentrations in blood serum than CA and CMCA NPs, demonstrating their long circulating characteristics. This is further supported by the low distribution of DOX in kidney delivered through α-KAMCA NPs after 24 h. The size of α-KAMCA NPs ranging from 230 nm to 360 nm reduced the probability of renal clearance, with surface-anchored α-KA circumventing opsonization and uptake by the reticuloendothelial system (RES), resulting in a prolonged circulation time for the drug. A very important aspect is the significant reduction in the distribution of DOX to the heart by CA, CMCA, and α-KAMCA NPs, indicating their potential in reducing the cardiotoxicity associated with DOX therapy.

A substantially lower amount of DOX-loaded CA, CMCA, and α-KAMCA NPs were detected in the kidneys compared to free DOX, which could be due to the enhancement of drug retention in plasma by NPs. The cutoff size for glomerular clearance of NPs is ~5 nm [66], indicating that free drug and very small particles used the kidney as an elimination route. We observed minimal accumulation in the major reticuloendothelial organs, including the lungs and spleen, which means that the NPs were masked from the RES.

Interestingly, the tumor accumulation of DOX was proportionate to the serum DOX level, while almost eight-fold more DOX accumulation was observed for DOX-loaded α-KAMCA NPs than free DOX at 2 h of treatment. Likewise, DOX-loaded CA NPs also showed a proportionate accumulation of drug in the tumor and blood serum, which exhibited almost five-fold more drug accumulation compared to the free form at 2 h of treatment. A similar finding was observed at 24 h of treatment for the same NPs. However, the free DOX accumulation in the heart was 4.28 ± 0.63 ng/mg, which was reduced to 2.73 ± 0.38 ng/mg, 0.59 ± 0.08 ng/mg, and 0.58 ± 0.02 ng/mg by incorporating DOX into CA, CMCA, and α-KAMCA NPs, respectively, at 24 h of treatment, shedding light on the potential application of CMCA and α-KAMCA NPs in avoiding DOX-associated cardiomyopathy. NPs with a diameter range from 100–150 nm have a tendency to circulate in the normal blood vessels and perfuse the capillaries of kidney, lung, and heart tissues [56,67]. In this study, the drug-loaded NPs possessed a size between 180 and 436 nm (Figure 9), which is bigger than the cut-off size for perfusing capillary tissues, especially in the heart, ensuring less accumulation of DOX-loaded NPs in the heart. However, the DOX–nanocarrier complex can easily pass through the leaky vasculature of the tumor region due to the bigger pore size of the tumor vessels, and it can accumulate for a longer period of time by virtue of the impaired lymphatic drainage in the tumor site (EPR effect), resulting in efficient uptake of drug-loaded NPs by target cancer cells with more antitumor efficacy and eventually less cytotoxicity.

The acidic microenvironment surrounding cancer cells contains specific types of proteins which might modify the composition of the protein corona around NPs, resulting in a change in bioavailability, as well as alteration of the interactions between the NPs and membranes of macrophages, endothelial cells, and target cells, and the overall therapeutic response [68]. The protein–NP interaction influences opsonization, circulation half-life, and cellular uptake. The protein corona can be characterized by total quantity, density, thickness, composition, relative abundance of each protein, protein binding affinity, and protein conformation. Complement factors, fibrinogen, or immunoglobulin (IgG), which are known as opsonins, would induce macrophage recognition and engulfment of NPs, leading to the fast clearance of NPs out of the body. In contrast, albumin or apolipoproteins (Apos), named dysopsonins, could extend the circulation time of the NPs [69,70,71,72]. This suggests that drug-loaded CA, CMCA, and α-KAMCA NPs are less likely to be cleared out by the reticuloendothelial system and, thus, would have an increased half-life in plasma, which can be further supported by the pharmacokinetics study of NP-treated mice. The α-KAMCA NPs resulted in higher DOX concentrations in blood serum than CA and CMCA NPs, demonstrating their long circulating characteristics due to the presence of dysopsonins on their protein corona. The size of α-KAMCA NPs ranging from 230 nm to 360 nm reduced the probability of reticuloendothelial system (RES) uptake via opsonization, resulting in a prolonged circulation time for the drug. The long circulation property of α-KAMCA NPs further enhanced DOX accumulation in the tumor, supported by the notion that NPs could enter into tumors through an EPR effect.

However, there are some limitations to using DOX fluorescence as a measure of concentration due to its self-quenching or the pH sensitivity of excitation or emission spectra [73], which was carefully taken care of during our experiment and data interpretation. Firstly, as our NPs are pH-sensitive, the pH of the lysis buffer was strictly maintained at 7.4 during the experiment to maintain consistency. Secondly, in our previous study, we used visualization by confocal microscopy to observe cellular uptake, as well as a cell viability test using different DOX-loaded NPs to correlate between the cellular uptake and cytotoxicity [28]. Considering all results, we found DOX as a potential model drug to measure fluorescence intensity in different organs and plasma to measure the concentration of free DOX and DOX-loaded NPs.

No definite signs of toxicity in mice treated with CA NPs, CMCA NPs, and α-KAMCA NPs were observed in acute and sub-chronic toxicity analysis (Table 4 and Table 5), which could be supported by the tumor regression study. The drug-free NPs showed no difference in tumor proliferation compared to the untreated mice group, pointing out that NPs are safe as a drug carrier without any toxicity.

## 4. Methods and Materials

### 4.1. Reagents

Dulbecco’s modified Eagle medium (DMEM) powder was purchased from Gibco by Life Technology (Thermo Fisher Scientific, USA). Calcium chloride dihydrate (CaCl_2_∙2H_2_O), sodium bicarbonate (NaHCO_3_), sodium hydrogen phosphate (Na_2_HPO_4_), potassium dihydrogen phosphate (KH_2_PO_4_), sodium citrate tribasic dihydrate, and alpha-ketoglutaric acid disodium salt hydrate were obtained from Sigma-Aldrich (St Louis, MO, USA). Anticancer drugs DOX and CYP were acquired from Sigma-Aldrich (St Louis, MO, USA). DMEM liquid media, fetal bovine serum (FBS), trypsin, TrypLE Express, penicillin–streptomycin, and trypan blue were procured from Sigma-Aldrich (St Louis, MO, USA). Sodium chloride and potassium chloride salts were bought from Fischer Scientific (Loughborough, UK). Acetonitrile (ACN), hydrochloric acid (HCl), and methanol were from Fischer Scientific (Loughborough, UK). Ammonium bicarbonate, formic acid, dithiothreitol (DTT), trifluoroacetic (TFA), trifluoroethanol (TFE) acid, and iodoacetamide (IAM) were from Sigma-Aldrich (St. Louis, MO, USA).

### 4.2. Synthesis and Encapsulation Efficiency of Drug-Loaded CA, CMCA, and α-KAMCA NPs

DMEM-buffered medium was prepared by mixing 0.675 g of DMEM powder with 44 mM sodium bicarbonate in Milli Q water, and the pH of the media was adjusted to 7.4 by using 0.1 M HCl. The final medium was filtered by using a sterilized non-pyrogenic 0.2-µm PES (polyethersulfone) membrane containing a syringe filter (Sartorius Stedim Biotech, Göttingen, Germany).

High-performance liquid chromatography (HPLC) was performed to assess CYP binding affinity with the NPs. CYP at 100 µM (26.11 µg/mL) concentration was added with 4 mM exogenous calcium to 1 mL of filtered DMEM medium to synthesize CYP-loaded CA NPs. Likewise, the same concentration of CYP was used along with 4 mM exogenous Ca^2+^ and 1 mM sodium citrate in 1 mL of filtered DMEM solution to generate CYP-bound CMCA NPs. To produce CYP-incorporated α-KAMCA NPs, CYP at 100 µM concentration was mixed with 4 mM exogenous Ca^2+^ and 4 mM alpha-ketoglutaric acid salt in 1 mL of filtered DMEM solution. All the salts and the drug mixture were then placed inside an incubator at 37 °C for 30 min to formulate drug-bound NPs. Next, the drug–particle suspension was centrifuged at 4 °C temperature for 30 min at 13,000 rpm to accelerate the particle sedimentation. The clear supernatant of the sedimented suspension was collected and run by using Agilent Chemostation software attached to HPLC (Agilent, Santa Clara, CA, USA) to check the amount of unbound drug present in the supernatant. This chromatography experiment was performed using a Zorbax C18 column (4.6 × 150 mm, Agilent, Santa Clara, CA, USA), and the mobile phase was ACN and Milli Q water in a proportion of 30:70 (*v*/*v*). The run time, diode array detection (DAD) wavelength, injection volume, and column temperature were fixed to 10 min, 197 nm, 20 µL, and 30 °C, respectively.

The standard curve was prepared by dissolving different concentrations of CYP from 0 µM to 100 µM (*y* = 7.9458*x* + 5.2171, *R^2^* = 0.9997) to calculate the concentrations of unbound CYP in the supernatant. $$N=\frac{P-5.2171\;}{7.9458}.$$
Q=M−N.

The percentage drug encapsulation efficiency was calculated using the following formula:$$\%\;Encapsulation\;efficiency=\frac{Q}{M}\;\times 100\%,$$ where *N* is the concentration of unbound drug or the volume of drug not entrapped by the NPs, *P* is the peak area of CYP present in the supernatant detected by HPLC, *Q* is the encapsulated concentration of CYP in the NPs, and *M* is the initial or total concentration of CYP used to formulate the CYP-bound NP formulations.

A standard curve was made by plotting different concentrations of DOX (0–15 µM) vs. fluorescence intensity. DOX at 10 µM (5.77 µg/mL) was added to CA NPs, CMCA NPs, and α-KAMCA NPs. After incubation for 30 min at 37 °C to form DOX-loaded CA, CMCA, and α-KAMCA NPs, the particle suspension was centrifuged twice at 6000 rpm for 20 min at 4 °C. Finally, after removing supernatant, the pellet was mixed with 10 mM ethylenediaminetetraacetic acid (EDTA) in phosphate-buffered saline (PBS) to dissolve the particles and release the bound drugs, and it was then subjected to a fluorescence intensity measurement with an excitation wavelength of 485 nm and an emission wavelength of 535 nm using Perkin Elmer 2030 manager software attached to a 2030 multilabel reader victor X5 (Perkin Elmer).

The concentrations of DOX present in the suspension were calculated from the fluorescence intensity, using the standard curve. The percentage binding efficiency was calculated using the following formula:$$\%\;Bnding\;affinity=\frac{\left[Z\right]NP\;bound\;drug}{\left[Z\right]initial}\times 100\%,$$ where [*Z*] *NP bound drug* is the fluorescence intensity of DOX-loaded NPs, and [*Z*] *initial* is the fluorescence intensity of DOX used to prepare the standard curve. The experiments were performed in triplicate, and the results were presented as means ± SD (standard deviation).

### 4.3. Morphology Analysis of CYP- and DOX-Loaded CA and α-KAMCA NPs by FE-SEM

CYP or DOX in 1 µM concentration was added along with 4 mM exogenous calcium and/or alpha-ketoglutaric acid and/or sodium citrate salt to 1 mL of filtered DMEM medium prior to incubation at 37 °C for 30 min to formulate CYP- or DOX-loaded NPs. The formulated drug-loaded NPs were centrifuged at 13,000 rpm for 15 min with a refrigerated bench-top microcentrifuge to collect the pellets. Then, 10 μL of each sample was withdrawn and placed onto a carbon tape-coated sample holder and dried at room temperature, followed by platinum sputtering of the dried samples with 30 mA sputter current at a 2.30 tooling factor for 40 s. The size and morphology of sputtered NPs were visualized at 5 kV using a field-emission scanning electron microscope (FE-SEM Hitachi/SU8010, Japan).

### 4.4. Murine Breast Cancer Cell Culture

Murine breast cancer-derived 4T1 cells were cultured in complete DMEM (cDMEM) medium (pH 7.4) containing 1% penicillin and streptomycin antibiotic and 10% FBS in a 25-cm^2^ flask, and the cell-containing flask was placed in a humidified incubator at 37 °C with 5% CO_2_. The cells were collected from the exponential growth phase and exposed to the trypsinization process, followed by repeated washing through centrifugation steps; they were then stained with trypan blue, counted using a hemocytometer, and then re-suspended in cDMEM medium at a concentration of 10^6^ cells/mL. From this cell stock medium, the calculated volume of cell-containing medium was mixed with 100 µL of PBS (pH 7.4) to ensure 10^5^ cells/100 µL for injecting in an animal model.

### 4.5. Tumor Regression Studies

Female Balb/c mice of 6–8 weeks old weighing 16–19 g were obtained from the School of Medicine and Health Science animal facility, Monash University. All animals were kept under 12-h/12-h light/dark conditions with free access to ab libitum and water. In vivo experiments were conducted in accordance with the approved animal ethics by the Monash Animal Ethics Committee (MARP/2016/126 and MUM/2018/12).

Approximately 1 × 10^5^ 4T1 cells/100 µL in PBS were subcutaneously injected on the mammary pad. Eleven days after inoculation, the mice were randomly divided into groups (*n* = 5) when the tumor volume reached an average of 13.63 ± 2.68 mm^3^. The length and width of an outgrowth tumor were estimated in mm by using a digital Vernier caliper over the period of 27 days, and the mice were sacrificed humanely via euthanasia, followed by measuring the weight of different organs of the mice. The body weights of mice with an induced tumor were measured in three-day intervals. The tumor volume data were presented as means ± SD of five different mice from each group. The following formula was used to calculate the tumor volume:$$Tumour\;volume\;\left({\mathrm{mm}}^{3}\right)=\frac{1}{2}\;\left(Length\times Width^{2}\right).$$

### 4.6. Fabrication of NPs and CYP-Loaded NPs for Tumor Model

Filtered DMEM medium was prepared by following the abovementioned protocol. CA NPs were generated by adding 40 mM exogenous Ca^2+^ in 100 µL of filtered serum-free DMEM medium. Similarly, 40 mM exogenous Ca^2+^ and 10 mM sodium citrate were mixed in 100 µL of serum-free DMEM to formulate CMCA NPs. The α-KAMCA NPs were produced by combining 40 mM Ca^2+^ and 40 mM α-ketoglutaric acid in 100 µL of filtered serum-free DMEM solution. CYP at 198.79 µM (equivalent to 0.3 mg/kg BW) concentration was added to DMEM-containing PCR tubes along with the appropriate amount of exogenous Ca^2+^ in the presence or absence of citrate or α-ketoglutaric acid to synthesize CYP-loaded NPs. Also, CYP at 198.79 µM concentration was dissolved in serum-free filtered DMEM medium as a control. All the different mixtures with or without the drug were placed in an incubator for 30 min at 37 °C. 

For the tumor regression study, one group of tumor-bearing mice was kept untreated as a negative control, the other tumor-bearing mice groups were treated with drug-free NPs or CYP only (dose of 0.3 mg/kg) as a positive control, and the remaining breast tumor-bearing mice were injected with CYP-loaded NPs (dose of 0.3 mg/kg) as a treatment group. The CYP solution, CYP-loaded CA NPs, CYP-loaded CMCA NPs, and CYP-loaded α-KAMCA NPs, as well as CA NPs, CMCA NPs, and α-KAMCA NPs, were injected intravenously using a 29G needle every two days with a total of three doses per mouse. The treatment was given at day 14 from tumor inoculation through the right or left caudal tail-vein, and the tumor volume was checked throughout the treatment period using a digital Vernier caliper.

### 4.7. Animal Biodistribution Studies

To investigate the tissue biodistribution of CA NPs, CMCA NPs, and α-KAMCA NPs, female Balb/c mice of 6–8 weeks old weighing 16–19 g were injected subcutaneously on the mammary pad with approximately 1 × 10^5^ 4T1 cells/100 µL in PBS. The mice were divided into groups (*n* = 5) when the tumor volume reached an average of 13.63 ± 2.68 mm^3^.

The DOX solution, DOX-loaded CA NPs, DOX-loaded CMCA NPs, and DOX-loaded α-KAMCA NPs were injected intravenously through the right or left caudal tail-vein with a DOX concentration of 5 mg/kg per dose. The mice were sacrificed after 2 h and 24 h of tail-vein injection (*n* = 5). The organs such as the liver, kidney, heart, spleen, lungs, and brain, as well as the tumor tissues, were isolated and kept in lysis buffer (pH 7.4) and kept at −150 °C in a freezer until further analysis. Additionally, the weights of all the organs and tumor tissues were measured for quantitative analysis of the presence of drug-loaded NPs.

Organs were homogenized and lysed with a lysis reagent (sodium fluoride and stable stock lysis buffer). The mixture was centrifuged at 8000 rpm at 4 °C for 50 min. The supernatant was collected in 1.5-mL microcentrifuge tubes. Then, 100 µL of supernatant was taken in black 96-well plates and subjected to fluorescence intensity measurement, quantified with an excitation wavelength of 485 nm and an emission wavelength of 535 nm using Perkin Elmer 2030 manager software attached to a 2030 multilabel reader victor ×5 (Perkin Elmer). The data were presented as means ± SD in relative fluorescence units (RFU)/mg of organ weight.

### 4.8. Blood Analysis

To determine the serum level of total drug-loaded NPs, blood samples of all the mice treated (*n* = 5) were collected by cardiac puncture using a 1-mL syringe with a 25G needle. A secondary method of euthanasia was performed after a cardiac puncture to sacrifice the animal. The blood samples were collected at two time points of analysis (2 h and 24 h of treatment). The blood samples were allowed to clot by leaving them at room temperature and then centrifuged at 10,000 rpm for 30 min at 4 °C to collect the supernatant serum. Afterward, 100 µL of each serum sample was collected and stored at −80 °C until it was assayed. The fluorescence emission of serum was measured using Perkin Elmer 2030 manager software attached to a 2030 multilabel reader victor X5 (Perkin Elmer) at 485/535 nm. The data were illustrated as means ± SD in RFU/100 µL of serum.

### 4.9. Toxicology Screening

According to the Organization for Economic Co-operation and Development (OECD) guidelines, the animals were screened for the toxicology study for the administration of different NPs [74]. To analyze sub-chronic toxicity and to observe the mice’s overall health and well-being, CA NPs (40 mM exogenous Ca^2+^ in 100 µL of filtered serum-free DMEM medium), CMCA NPs (40 mM exogenous Ca^2+^ and 10 mM sodium citrate in 100 µL of serum-free DMEM), and α-KAMCA NPs (40 mM Ca^2+^ and 40 mM α-ketoglutaric acid in in 100 µL of filtered serum-free DMEM solution) were administered intravenously through the right or left caudal tail-vein three times at two-day intervals, and the blood was collected by cardiac puncture after 27 days of the first treatment. For acute toxicity analysis, mice were treated intravenously with CA NPs (40 mM exogenous Ca^2+^ in 100 µL of filtered serum-free DMEM media), CMCA NPs (40 mM exogenous Ca^2+^ and 10 mM sodium citrate in 100 µL of serum-free DMEM), and α-KAMCA NPs (40 mM Ca^2+^ and 40 mM α-ketoglutaric acid in 100 µL of filtered serum-free DMEM solution), and the blood sample was collected by cardiac puncture after 24 h of treatment.

During the sub-chronic study, the animals were well taken care throughout the 28 days, and any type of abnormality in behavior (e.g., less intake of water, less food intake, weight loss, less mobility, feces color change) of the treated mice was monitored. At each specific time point, the maximum volume of blood was extracted by cardiac puncture, and the mice were sacrificed humanely via euthanasia. The blood was collected into lithium heparin tubes (Vacutube, Vodice, Slovenia) and later transported in an ice box maintained at 4 °C to the Hematology and Biochemistry Clinical Laboratory, Faculty of Veterinary, University Putra Malaysia.

The test included the assessment of biomarkers for a blood electrolyte panel (Na^+^, K^+^, Cl^−^, Ca^2+^, PO_4_^3−^), toxicity in the liver (i.e., alanine transaminase (ALT), alkaline phosphatase (ALP), total bilirubin (TBil), total protein (TP), and albumin (ALB) levels), kidney (i.e., urea level), and pancreas (i.e., amylase (Amy) level). All values and findings were compared between treated and untreated groups.

### 4.10. Protein Corona Analysis Using Liquid Chromatography and Mass Spectrometry (LC/MS)

CA NPs were prepared by adding 4 mM exogenous Ca^2+^ with incubation at 37 °C for 30 min. Similarly, CMCA and α-KAMCA NPs were synthesized by adding 1 mM sodium citrate and 4 mM α-ketoglutarate, respectively, along with 4 mM exogenous Ca^2+^, prior to incubation at 37 °C for 30 min. Then, 10% FBS was added to the particle suspensions before incubation at 37 °C for 10 min. The samples were then centrifuged at 13,000 rpm and 20 °C for 10 min. The supernatant was then removed and replaced with Milli Q water without mixing with the precipitate, and it was subjected to centrifugation under the same setting. The precipitate was collected and resuspended in 100 mL of 50 mM EDTA.

A C18 spin column (Thermo Fisher Scientific, USA) was used to perform liquid chromatography. The column was centrifuged at 1500× *g* relative centrifugal force (rcf) for 1 min, followed by discarding flow-through and repetition of a similar process. Then, 200 µL of equilibration solution (0.5% TFA in 5% ACN) was added and centrifuged at 1500× *g* rcf for 1 min. The NP samples were mixed with 1/3 sample buffer (2% TFA in 20% ACN) and loaded on top of the resin bed. The spin column was placed into a receiver and again centrifuged at 1500× *g* rcf for 1 min. Afterward, 200 µL of wash solution was run through the column, followed by the addition of 20 µL of elution buffer (70% ACN). After centrifugation, the collected flow-through (40 L for each sample) was dried in a Labogene vacuum evaporator (Bjarkesvej, Lillerød, Denmark) at 1000 rpm for 3 h at 50 °C.

### 4.11. In-Solution Digestion

The pellet formed from the evaporated sample was resuspended in 100 µL. Then, 50 μL was drawn out for further experimentation. It was added to 25 μL of ammonium bicarbonate, 25 µL of TFE, and 1 µL of DTT stock. The sample was then incubated at 60 °C for 60 min, and 4 µL of IAM was added; the sample was kept at room temperature in dark conditions for 60 min. The samples were mixed with 300 µL of water, 100 µL of ammonium bicarbonate, and 5 µL of trypsin, and they were incubated at 37 °C overnight. The prepared samples were subjected to mass spectroscopy Agilent MassHunter (Agilent Technologies), and the results were analyzed by protein identification through automated de novo sequencing using PEAKS Studio 7.0 (Bioinformatics Solution Inc.).

### 4.12. Dissolution Study of CA, CMCA, and α-KAMCA NPs at Different pH

CA NPs were generated using 20 mM Ca^2+^, CMCA NPs were prepared with 20 mM Ca^2+^ and 5 mM sodium citrate, and α-KAMCA NPs were synthesized using 20 mM Ca^2+^ and 20 mM alpha-ketoglutarate in 200-µL PCR tubes. All the salt mixtures in different PCR tubes were incubated at 37 °C for 30 min to formulate the corresponding NPs. After this, all the formulated NPs were suspended into 800 µL of DMEM medium at different pH (7.5, 7.0, 6.5, 6.0, 5.5, and 5.0), prior to the measurement of absorbance at 320 nm using an ultraviolet–visible light (UV–Vis) spectrophotometer. Data were represented as means ± SD of triplicates.

### 4.13. Statistical Analysis

Statistical significance was analyzed in drug-treated mice group versus drug-loaded NP-treated groups by one-way ANOVA (analysis of variance), followed by post hoc analyses using a Tukey multiple comparisons test with SPSS v23 for Windows. The minimal level of statistical significance was *p* < 0.05 with a 95% confidence interval (CI). The experimental data were presented as means ± SD (*n* = 5).

## 5. Conclusions

In conclusion, CMCA NPs and α-KAMCA NPs exhibit promising roles as nanocarriers for therapeutics, prolonging the plasma half-life and improving tumor accumulation of the encapsulated drug, thus drastically inhibiting tumor growth in a syngeneic mouse model of breast cancer. This study also demonstrated significantly reduced drug availability in healthy organs and tissues with no significant difference in blood biochemistry in NP-treated mice in comparison to untreated ones.

## Figures and Tables

**Figure 1 cancers-12-00161-f001:**
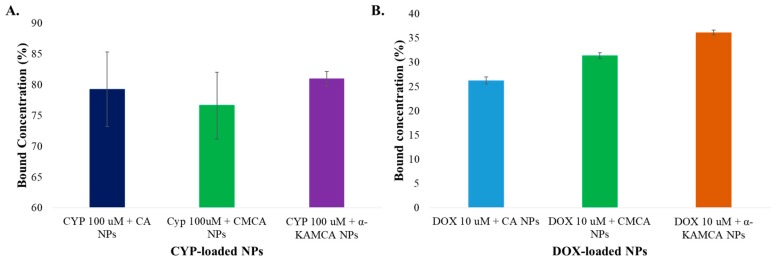
Drug binding affinity for carbonate apatite (CA) nanoparticles (NPs), citrate-modified CA (CMCA) NPs, and α-ketoglutaric acid-modified CA (α-KAMCA) NPs. (**A**) Estimation of cyclophosphamide (CYP) binding affinity for CA NPs, CMCA NPs, and α-KAMCA NPs. (**B**) Estimation of doxorubicin (DOX) binding affinity for CA NPs, CMCA NPs, and α-KAMCA NPs.

**Figure 2 cancers-12-00161-f002:**
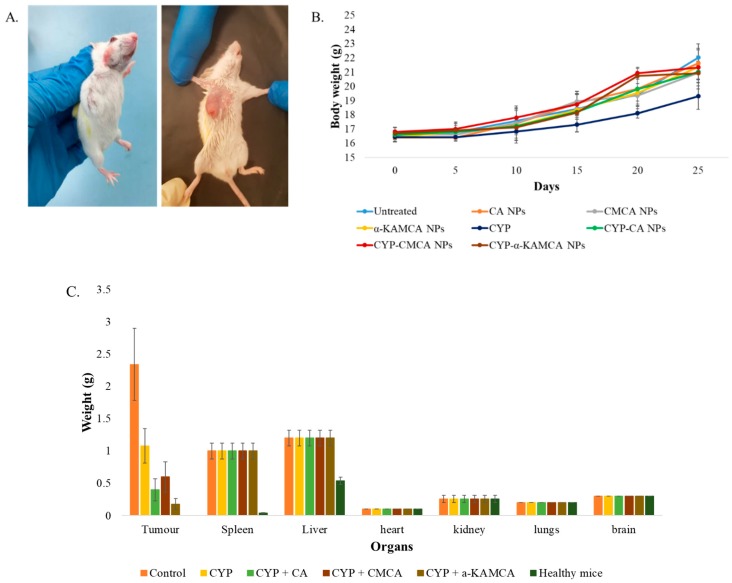
Breast cancer mouse model. (**A**) Murine breast cancer model established by injecting 4T1 cells subcutaneously on the mammary pad of mice (*n* = 5). (**B**) Body weight (BW) curves for 25 days during tumor inoculation and treatment. (**C**) Different organ weights of healthy, untreated, free CYP and CYP-loaded NP-treated mice groups at day 27 after the mice were sacrificed humanely via euthanasia.

**Figure 3 cancers-12-00161-f003:**
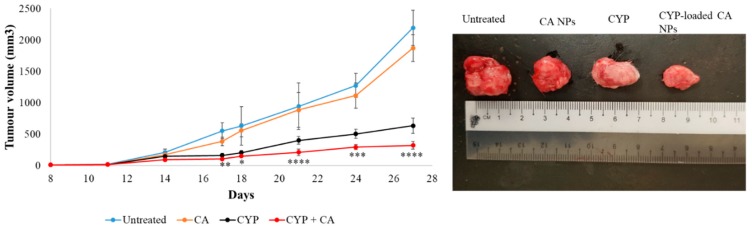
Treatment effects of free CYP, CYP-loaded CA NPs, and drug-free CA NPs compared to the untreated group of mice. Tumor picture was taken at day 27 after the mice were sacrificed humanely via euthanasia. Values were significant (*) at a *p*-value of 0.01 to 0.05, very significant (**) at a *p*-value of 0.001 to 0.01, highly significant (***) at a *p*-value of 0.0001 to 0.001, and extremely significant (****) at a *p*-value < 0.0001 vs. the same treatment with free CYP at a confidence interval (CI) of 95%.

**Figure 4 cancers-12-00161-f004:**
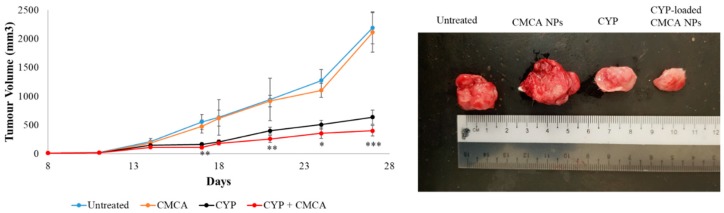
Treatment effects of free CYP, CYP-loaded CMCA NPs, and drug-free CMCA NPs compared to the untreated group of mice. Tumor picture was taken at day 27 after the mice were sacrificed humanely via euthanasia. Values were significant (*) at a *p*-value of 0.01 to 0.05, very significant (**) at a *p*-value of 0.001 to 0.01, and highly significant (***) at a *p*-value of 0.0001 to 0.001 vs. the same treatment with free CYP at a CI of 95%.

**Figure 5 cancers-12-00161-f005:**
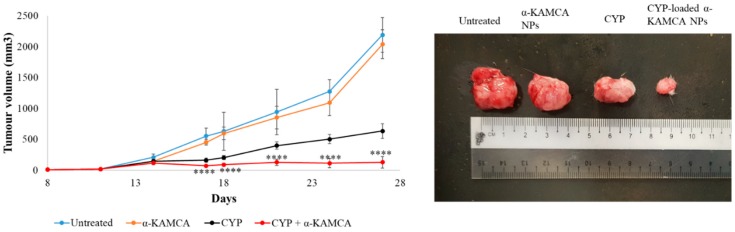
Treatment effects of free CYP, CYP-loaded α-KAMCA NPs, and drug-free α-KAMCA NPs compared to the untreated group of mice. Tumor picture was taken at day 27 after the mice were sacrificed humanely via euthanasia. Values were extremely significant (****) at a *p*-value < 0.0001 vs. the same treatment with free CYP at a CI of 95%.

**Figure 6 cancers-12-00161-f006:**
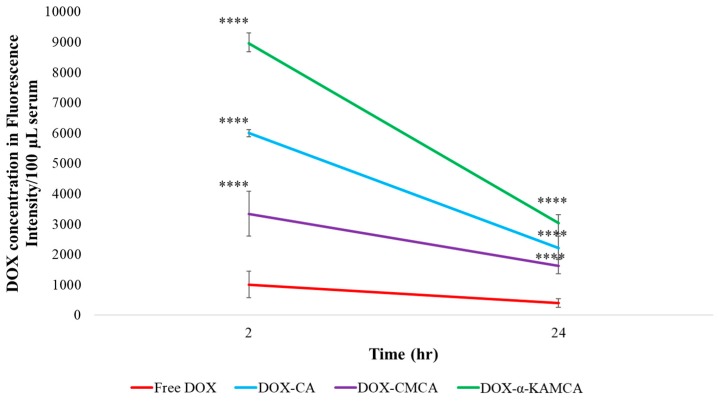
Experimental schema of the circulation time of DOX-loaded CA NPs, CMCA NPs, and α-KAMCA NPs compared to free DOX following an intravenous injection. Values were extremely significant (****) at a *p*-value < 0.0001 vs. the same treatment with free DOX at a CI of 95%.

**Figure 7 cancers-12-00161-f007:**
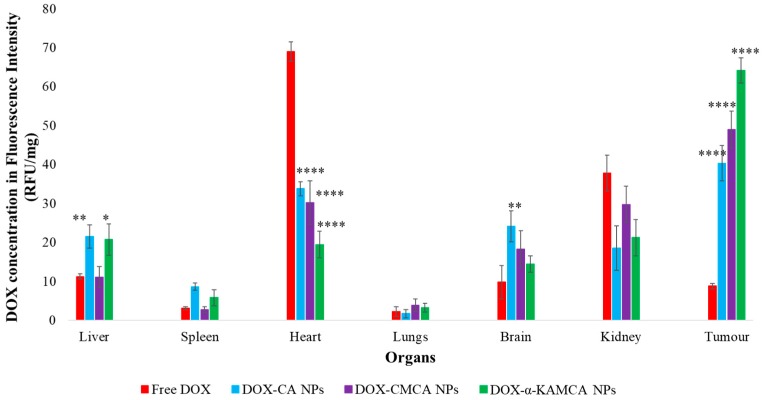
Biodistribution of free DOX, DOX-loaded CA NPs, DOX-loaded CMCA NPs, and DOX-loaded α-KAMCA NPs in the 4T1 murine breast cancer cell line-induced tumor mouse model at 2 h of treatment. Values were significant (*) at a *p*-value of 0.01 to 0.05, very significant (**) at a *p*-value of 0.001 to 0.01, and extremely significant (****) at a *p*-value < 0.0001 vs. the same treatment with free DOX at a CI of 95%.

**Figure 8 cancers-12-00161-f008:**
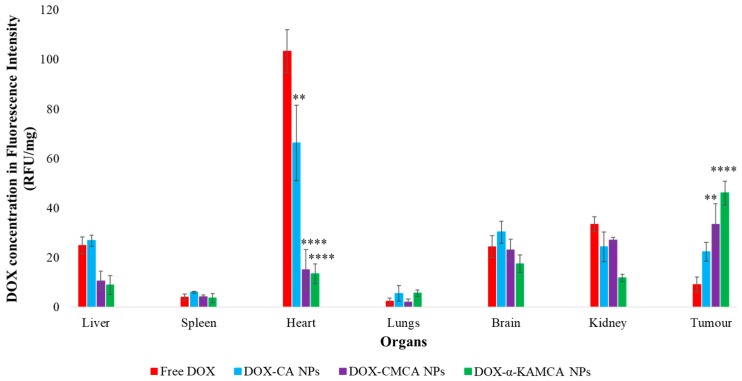
Biodistribution of free DOX, DOX-loaded CA NPs, DOX-loaded CMCA NPs, and DOX-loaded α-KAMCA NPs in the 4T1 murine breast cancer cell line-induced tumor mouse model at 24 h of treatment. Values were significant (*) at a *p*-value of 0.01 to 0.05, very significant (**) at a *p*-value of 0.001 to 0.01, and extremely significant (****) at a *p*-value <0.0001 vs. the same treatment with free DOX at a CI of 95%.

**Figure 9 cancers-12-00161-f009:**
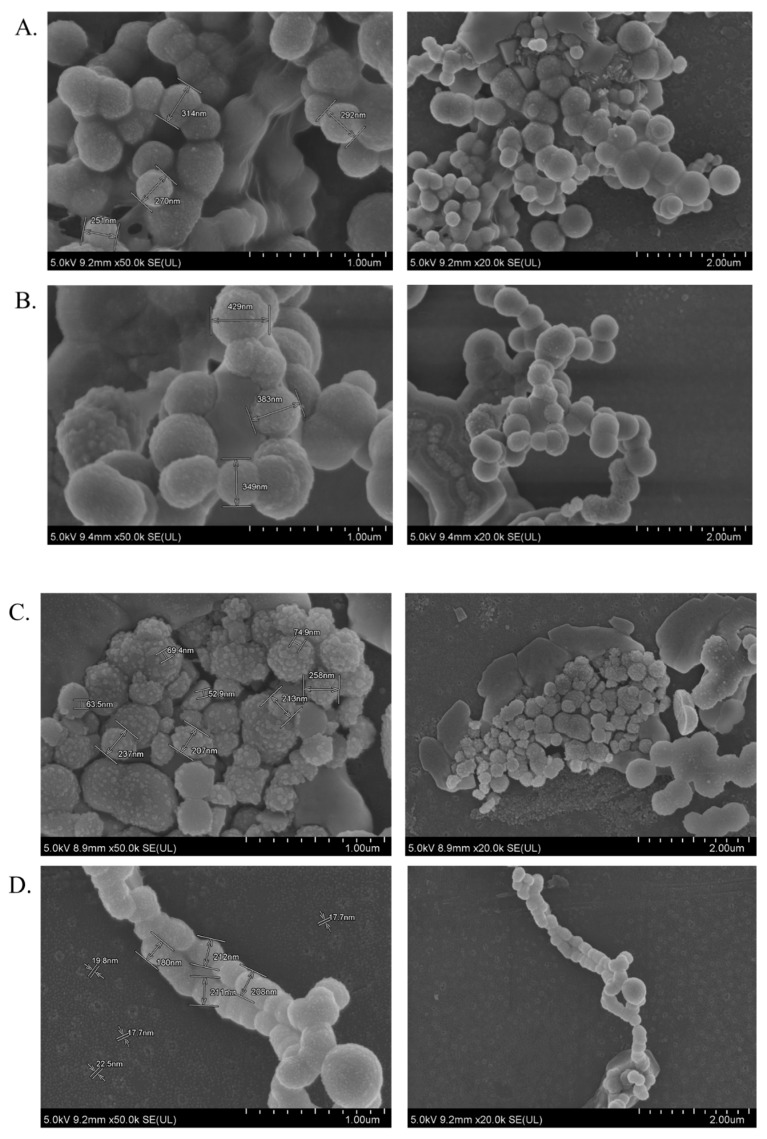
Morphology analysis of drug-loaded CA and α-KAMCA NPs by field emission (FE)-SEM: (**A**) CYP-loaded CA NPs; (**B**) DOX-loaded CA NPs; (**C**) CYP-loaded α-KAMCA NPs; (**D**) DOX-loaded α-KAMCA NPs; (**E**) CYP-loaded CMCA NPs; (**F**) DOX-loaded CMCA NPs.

**Figure 10 cancers-12-00161-f010:**
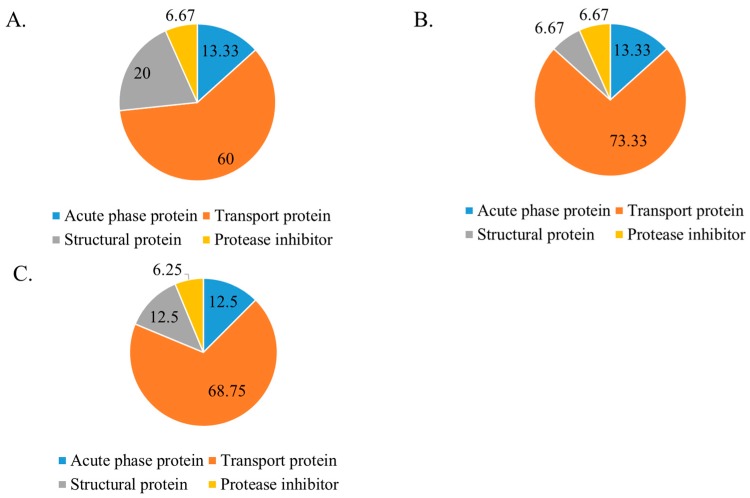
Protein distribution (%) around—(**A**) CA NPs in 10% fetal bovine serum (FBS), (**B**) CMCA NPs in 10% FBS, and (**C**) α-KAMCA NPs in 10% FBS.

**Figure 11 cancers-12-00161-f011:**
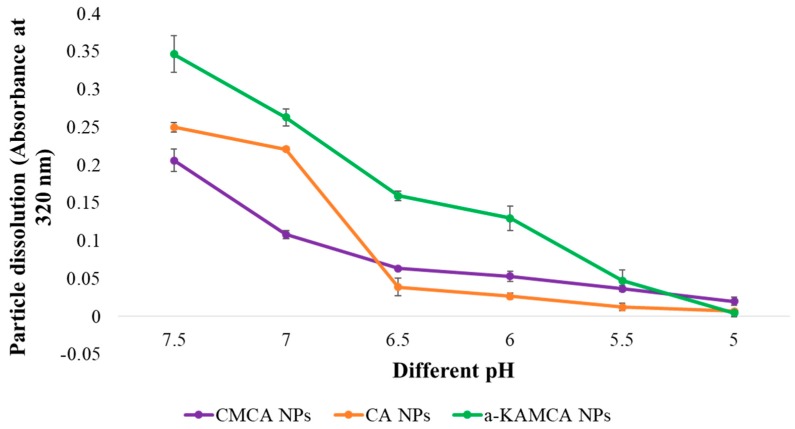
The pH-dependent dissolution study of CA, CMCA, and α-KAMCA NPs.

**Table 1 cancers-12-00161-t001:** Estimation of serum drug concentration following an intravenous injection of free doxorubicin (DOX), DOX-loaded carbonate apatite (CA) nanoparticles (NPs), DOX-loaded citrate-modified CA (CMCA) NPs, and DOX-loaded α-ketoglutaric acid-modified CA (α-KAMCA) NPs (*n* = 5).

Formulations	2 h (ng/100 μL)	24 h (ng/100 μL)
Free DOX	110.53 ± 11.83	93.74 ± 3.89
DOX–CA	538.13 ± 37.01	324.75 ± 19.79
DOX–CMCA	339.01 ± 19.64	237.06 ± 27.02
DOX–α-KAMCA	877.21 ± 9.33	440.28 ± 7.78

**Table 2 cancers-12-00161-t002:** Organ distribution of free DOX, DOX-loaded CA NPs, DOX-loaded CMCA NPs, and DOX-loaded α-KAMCA NPs at 2 h of treatment (*n* = 5).

Organs	Free DOX (ng/mg)	DOX–CA (ng/mg)	DOX–CMCA (ng/mg)	DOX–α-KAMCA (ng/mg)
Liver	0.3 ± 0.01	0.92 ± 0.04	0.3 ± 0.01	0.82 ± 0.11
Spleen	0.13 ± 0.01	0.22 ± 0.01	0.13 ± 0.01	0.21 ± 0.02
Heart	2.86 ± 0.23	1.39 ± 0.12	1.16 ± 0.08	0.77 ± 0.02
Lungs	0.11 ± 0.01	0.12 ± 0.01	0.14 ± 0.01	0.14 ± 0.01
Brain	0.29 ± 0.02	0.94 ± 0.01	0.64 ± 0.06	0.55 ± 0.04
Kidney	1.45 ± 0.01	0.69 ± 0.05	1.27 ± 0.07	0.77 ± 0.08
Tumor	0.37 ± 0.01	1.64 ± 0.06	2 ± 0.01	2.75 ± 0.31

**Table 3 cancers-12-00161-t003:** Organ distribution of free DOX, DOX-loaded CA NPs, DOX-loaded CMCA NPs, and DOX-loaded α-KAMCA NPs at 24 h of treatment (*n* = 5).

Organs	Free DOX (ng/mg)	DOX–CA (ng/mg)	DOX–CMCA (ng/mg)	DOX–α-KAMCA (ng/mg)
Liver	1.18 ± 0.31	1.01 ± 0.14	0.3 ± 0.01	0.31 ± 0.01
Spleen	0.14 ± 0.01	0.23 ± 0.01	0.14 ± 0.01	0.13 ± 0.01
Heart	4.28 ± 0.63	2.73 ± 0.38	0.59 ± 0.08	0.58 ± 0.02
Lungs	0.14 ± 0.01	0.25 ± 0.01	0.14 ± 0.01	0.1 ± 0.01
Brain	1.06 ± 0.02	1.35 ± 0.35	0.9 ± 0.06	0.55 ± 0.04
Kidney	1.45 ± 0.01	1.12 ± 0.05	1.27 ± 0.07	0.57 ± 0.01
Tumor	0.37 ± 0.01	1.04 ± 0.06	1.34 ± 0.1	1.79 ± 0.09

**Table 4 cancers-12-00161-t004:** Acute toxicity study in mice treated with CA, CMCA, and α-KAMCA NPs (*n* = 5). Phos—phosphate; ALP—alkaline phosphatase; TP—total protein; ALB—albumin; TBil—total bilirubin; Amy—amylase; ALT—alanine transaminase.

Blood Chemistry Test	Untreated Mice	CA NPs		CMCA NPs		α-KAMCA NPs	
	Mean ± SD	Mean ± SD	*p*-Value	Mean ± SD	*p*-Value	Mean ± SD	*p*-Value
Na^+^ (mmol/L)	149.8 ± 1.92	150.8 ± 2.49	Not significant	155.8 ± 0.84	0.003	160.8 ± 2.95	0.001
K^+^ (mmol/L)	4.16 ± 0.3	4.74 ± 0.63	Not significant	3.88 ± 0.28	Not significant	4.44 ± 0.23	Not significant
Cl^−^ (mmol/L)	113.6 ± 1.52	114 ± 3.94	Not significant	119.8 ± 1.30	0.007	123.2 ± 2.59	0.001
Ca^2+^ (mmol/L)	2.04 ± 0.11	2.18 ± 0.43	Not significant	2.5 ± 0.96	Not significant	2.3 ± 0.28	Not significant
Phos (mmol/L)	2.38 ± 0.13	4.16 ± 1.1	0.003	2.56 ± 0.52	Not significant	3.46 ± 0.51	Not significant
ALP (U/L)	82.8 ± 8.53	84.6 ± 7.67	Not significant	76.6 ± 9.6	Not significant	87 ± 10.37	Not significant
TP (g/L)	43.72 ± 2.29	48.02 ± 4.42	Not significant	45.4 ± 1.89	Not significant	50.94 ± 3.89	Not significant
ALB (g/L)	25.12 ± 1.6	26.86 ± 2.7	Not significant	26.2 ± 2.24	Not significant	27.62 ± 4.42	Not significant
TBil (µmol/L)	5.2 ± 0.35	4.56 ± 0.55	Not significant	5.86 ± 1.68	Not significant	4.74 ± 0.42	Not significant
Urea (mmol/L)	9.52 ± 1.68	11.44 ± 2.46	Not significant	10.84 ± 1.03	Not significant	14.98 ± 1.82	0.01
Amy (U/L)	1532 ± 85.84	1601.4 ± 160.98	Not significant	1475.8 ± 70.12	Not significant	1708 ± 151.49	Not significant
ALT (U/L)	22.8 ± 3.19	32 ± 10.95	Not significant	25 ± 4.47	Not significant	30 ± 5.5	Not significant

**Table 5 cancers-12-00161-t005:** Sub-chronic toxicity study in mice treated with CA, CMCA, and α-KAMCA NPs (*n* = 5).

Blood Chemistry Test	Untreated Mice	CA NPs		CMCA NPs		α-KAMCA NPs	
	Mean ± SD	Mean ± SD	*p*-Value	Mean ± SD	*p*-Value	Mean ± SD	*p*-Value
Na^+^ (mmol/L)	149.8 ± 1.92	150.8 ± 2.49	Not significant	155.8 ± 0.84	Not significant	160.8 ± 2.95	Not significant
K^+^ (mmol/L)	4.16 ± 0.3	4.74 ± 0.63	Not significant	3.88 ± 0.28	Not significant	4.44 ± 0.23	Not significant
Cl^−^ (mmol/L)	113.6 ± 1.52	114 ± 3.94	Not significant	119.8 ± 1.30	Not significant	123.2 ± 2.59	Not significant
Ca^2+^ (mmol/L)	2.04 ± 0.11	2.18 ± 0.43	Not significant	2.5 ± 0.96	Not significant	2.3 ± 0.28	0.05
Phos (mmol/L)	2.38 ± 0.13	4.16 ± 1.1	Not significant	2.56 ± 0.52	Not significant	3.46 ± 0.51	Not significant
ALP (U/L)	82.8 ± 8.53	84.6 ± 7.67	Not significant	76.6 ± 9.6	Not significant	87 ± 10.37	Not significant
TP (g/L)	43.72 ± 2.29	48.02 ± 4.42	Not significant	45.4 ± 1.89	Not significant	50.94 ± 3.89	Not significant
ALB (g/L)	25.12 ± 1.6	26.86 ± 2.7	Not significant	26.2 ± 2.24	Not significant	27.62 ± 4.42	Not significant
TBil (µmol/L)	5.2 ± 0.35	4.56 ± 0.55	Not significant	5.86 ± 1.68	Not significant	4.74 ± 0.42	Not significant
Urea (mmol/L)	9.52 ± 1.68	11.44 ± 2.46	Not significant	10.84 ± 1.03	Not significant	14.98 ± 1.82	Not significant
Amy (U/L)	1532 ± 85.84	1601.4 ± 160.98	0.001	1475.8 ± 70.12	Not significant	1708 ± 151.49	Not significant
ALT (U/L)	22.8 ± 3.19	32 ± 10.95	Not significant	25 ± 4.47	0.02	30 ± 5.5	Not significant

**Table 6 cancers-12-00161-t006:** Protein distribution around CA, CMCA, and α-KAMCA NPs in 10% fetal bovine serum (FBS).

	CA NPs			CMCA NPs			α-KAMCA NPs		
SL No.	Description	Coverage (%)	Unique/Peptides	Description	Coverage (%)	Unique/Peptides	Description	Coverage (%)	Unique/Peptides
1	ALB protein	76	7/49	ALB protein	80	7/50	ALB protein	82	6/55
2	Serum albumin	71	3/45	Serum albumin	78	6/49	Serum albumin	78	3/52
3	Serum albumin	71	3/45	Serum albumin	78	6/49	Serum albumin	78	3/52
4	Alpha-2-HS-glycoprotein	46	14/14	Alpha-2-HS-glycoprotein	55	19/19	Alpha-2-HS-glycoprotein	63	22/22
5	Alpha-2-HS-glycoprotein	46	14/14	Alpha-2-HS-glycoprotein	55	19/19	Alpha-2-HS-glycoprotein	63	22/22
6	Serotransferrin	21	12/12	Serotransferrin	32	16/17	Serotransferrin	14	9/9
7	Alpha-1-antiproteinase	21	6/6	Serotransferrin	30	1516	Serotransferrin	14	9/9
8	Alpha-fetoprotein	8	3/4	Alpha-1-antiproteinase	16	6/6	Alpha-1-antiproteinase	20	8/8
9	Vitamin D-binding protein	7	2/2	Alpha-fetoprotein	5	2/3	Keratin, type II cytoskeletal 7	5	2/2
10	Vitamin D-binding protein	7	2/2	Vitamin D-binding protein	8	3/3	Apolipoprotein A-II	17	2/2
11	Vitamin D-binding protein	7	2/2	Vitamin D-binding protein	8	3/3	Uncharacterized protein	19	2/2
12	Keratin, type II cytoskeletal 7	5	2/2	Vitamin D-binding protein	8	3/3	Hemoglobin fetal subunit beta	15	2/2
13	Keratin 1	6	3/3	Keratin, type II cytoskeletal 7	5	2/2	Globin A1	15	2/2
14	Keratin 10 (epidermolytic hyperkeratosis; keratosis palmaris et plantaris)	11	5/5	Globin C1	17	2/2	Hemoglobin subunit beta	15	2/2
15	Uncharacterized protein	19	2/2	Hemoglobin subunit alpha	17	2/2	Hemoglobin beta	15	2/2
16							Keratin 1	3	2/2
**Total Protein 16**									

**Table 7 cancers-12-00161-t007:** List of identified proteins around CA, CMCA, and α-KAMCA NPs in 10% FBS.

SL No.	Identified Protein	MW (Da)	Function	Protein Phase
1	ALB protein	69,293	Transport, binding protein	Transport protein
2	Serum albumin	69,323	Lipid binding, metal binding, transport	Transport protein
4	Alpha-2-HS-glycoprotein	38,419	Acute phase	Acute phase protein
6	Serotransferrin	77,738	Iron transport	Transport protein
7	Alpha-1-antiproteinase	46,104	Protease inhibitor	Protease inhibitor
8	Alpha-fetoprotein	68,588	Binds copper, nickel, and fatty acids, as well as bilirubin and serum albumin	Transport protein
9	Vitamin D-binding protein	53,328	Vitamin D transport and storage, scavenging of extracellular G-actin, enhancement of the chemotactic activity of C5 alpha for neutrophils in inflammation and macrophage activation	Transport protein
12	Keratin, type II cytoskeletal 7	51,578	Blocks interferon-dependent interphase and stimulates DNA synthesis in cells	Structural protein
13	Keratin 1	63,165	Carbohydrate binding, protein heterodimerization activity, structural constituent of epidermis, peptide cross-linking, protein heterotetramerization	Structural protein
14	Keratin 10 (epidermolytic hyperkeratosis; keratosis palmaris et plantaris)	54,849	Keratinocyte differentiation, peptide cross-linking, protein heterotetramerization	Structural protein
15	Uncharacterized protein	15,806	Metal binding	Transport protein
16	Apolipoprotein A-II	11,202	Stabilizes HDL (high-density lipoprotein) structure via its association with lipids, and affects HDL metabolism; has antimicrobial activity	Transport protein
18	Hemoglobin fetal subunit beta	15,859	Oxygen transport from the lung to the various peripheral tissues	Transport protein
19	Globin A1	15,954	Heme binding, metal ion binding, oxygen binding, oxygen carrier activity	Transport protein
20	Hemoglobin subunit beta	15,954	Oxygen transport from the lung to the various peripheral tissues	Transport protein
21	Hemoglobin beta	15,979	Heme binding, metal ion binding, oxygen binding, oxygen carrier activity	Transport protein
22	Globin C1	15,184	Heme binding, iron ion binding, oxygen binding, oxygen carrier activity	Transport protein

**Table 8 cancers-12-00161-t008:** Protein distribution (%) around CA NPs in 10% FBS.

SL No	Identified Protein	Distribution (%)	Protein Phase	Distribution (%)
1	ALB protein	6.67	Transport protein	60
2	Serum albumin	13.33	Transport protein	
3	Serum albumin		Transport protein	
4	Serotransferrin	6.67	Transport protein	
5	Alpha-fetoprotein	6.67	Transport protein	
6	Vitamin D-binding protein	20	Transport protein	
7	Vitamin D-binding protein		Transport protein	
8	Vitamin D-binding protein		Transport protein	
9	Uncharacterized protein	6.67	Transport protein	
10	Keratin, type II cytoskeletal 7	6.67	Structural protein	20
11	Keratin 1	6.67	Structural protein	
12	Keratin 10 (epidermolytic hyperkeratosis; keratosis palmaris et plantaris)	6.67	Structural protein	
13	Alpha-2-HS-glycoprotein	13.33	Acute phase protein	13.33
14	Alpha-2-HS-glycoprotein		Acute phase protein	
15	Alpha-1-antiproteinase	6.67	Protease inhibitor	6.67

**Table 9 cancers-12-00161-t009:** Protein distribution (%) around CMCA NPs in 10% FBS.

SL No	Identified Protein	Distribution (%)	Protein Phase	Distribution (%)
1	ALB protein	6.67	Transport protein	73.33
2	Serum albumin	13.33	Transport protein	
3	Serum albumin		Transport protein	
4	Hemoglobin subunit alpha	6.67	Transport protein	
5	Alpha-fetoprotein	6.67	Transport protein	
6	Serotransferrin	13.33	Transport protein	
7	Serotransferrin		Transport protein	
8	Globin C1	6.67	Transport protein	
9	Vitamin D-binding protein	20	Transport protein	
10	Vitamin D-binding protein		Transport protein	
11	Vitamin D-binding protein		Transport protein	
12	Alpha-2-HS-glycoprotein	13.33	Acute phase protein	13.33
13	Alpha-2-HS-glycoprotein		Acute phase protein	
14	Keratin, type II cytoskeletal 7	6.67	Structural protein	6.67
15	Alpha-1-antiproteinase	6.67	Protease inhibitor	6.67

**Table 10 cancers-12-00161-t010:** Protein distribution (%) around α-KAMCA NPs in 10% FBS.

SL No	Identified Protein	Distribution (%)	Protein Phase	Distribution (%)
1	ALB protein	6.25	Transport protein	68.75
2	Serum albumin	12.5	Transport protein	
3	Serum albumin		Transport protein	
4	Serotransferrin	12.5	Transport protein	
5	Serotransferrin		Transport protein	
6	Apolipoprotein A-II	6.25	Transport protein	
7	Uncharacterized protein	6.25	Transport protein	
8	Hemoglobin fetal subunit beta	6.25	Transport protein	
9	Hemoglobin beta	6.25	Transport protein	
10	Hemoglobin subunit beta	6.25	Transport protein	
11	Globin A1	6.25	Transport protein	
12	Keratin 1	6.25	Structural protein	12.5
13	Keratin, type II cytoskeletal 7	6.25	Structural protein	
14	Alpha-2-HS-glycoprotein	12.5	Acute phase protein	12.5
15	Alpha-2-HS-glycoprotein		Acute phase protein	
16	Alpha-1-antiproteinase	6.25	Protease inhibitor	6.25

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
