# Peer review of "Krebs Cycle Intermediate-Modified Carbonate Apatite Nanoparticles Drastically Reduce Mouse Tumor Burden and Toxicity by Restricting Broad Tissue Distribution of Anticancer Drugs"

_cancers, 2020, doi:10.3390/cancers12010161_

Round 1

Reviewer 1 Report

The authors demonstrated the citrate-modified carbonate apatite nanoparticles (CMCA NPs) and alpha-ketoglutaric acid-modified CA (alpha-KAMCA) NPs exhibited promising roles as nanocarriers for treatment of breast cancer.  The manuscript is well written, the aim of the study may be sound.  However, it is necessary to correct various problems before publication in Cancers.

It may be best to include the results of doxorubicin (DOX) binding affinity to CA, CMCA and alpha-KAMCA NPs. Authors should describe the legend of figures 3, 4 and 5 in the detail. I didn't know when the picture was taken after administration. In figures 6, 7 and 8, the authors should show the vertical axis in terms of DOX concentration, but not fluorescence intensity. It may be best to add the data of biodistribution (tissues and tumor) of cyclophosphamide (CYP) following administration of CYP-loaded CMCA NPs and CYP-loaded alpha-KAMCA NPs to the manuscript. Authors should add the data of weight of various tissues (liver, spleen, heart, lungs, brain, kidney) following administration of CYP- and DOX-loaded CMCA NPs and CYP- and DOX-loaded alpha-KAMCA NPs to the manuscript. Authors should add the data of markers with cardiac function as an index such as AST, ALT, CK-MB after administration of free DOX, DOX-loaded CMCA NPs and DOX-loaded alpha-KAMCA NPs to the manuscript, in order to support your data in figures 7 and 8. Authors should add the data of morphology analysis of CYP-loaded and DOX-loaded CMCA NPs to the manuscript. In line 275-279, the authors should explain the contents of Tables 6 and 7 in the more detail. Authors demonstrated that the accumulation of DOX in cancer cells was much higher in the CA-, CMCA- and alpha-KAMCA groups than in the free drug group (figures 7 and 8). On the other hand, the accumulation of DOX in heart was much lower in the CA-, CMCA- and alpha-KAMCA groups than in the free drug group (figures 7 and 8).  The authors should clearly explain why these phenomenon occurred in the Discussion section. In 496-498, the authors measured the concentration of DOX by detecting fluorescence sensitivity with a multilabel reader. The environment of NPs varies depending on the heterogeneous surface charge et al. Therefore, I thought that the detection of DOX fluorescence sensitivity might be affected by the external environment.  The authors should include using references that the fluorescence sensitivity of DOX is not affected by the external environment. The authors should correct careless mistakes (e.g. line 36, line 39).

Author Response

Answer:

Doxorubicin (DOX) binding affinity to CA, CMCA and α-KAMCA NPs has been added to the text (line 106 – 107) with the figure 1B.

The legends of figures 3, 4 and 5 have been described in the detail with time point.

DOX concentration in ng/mg has been presented in Table 1, 2 and 3. Also %DOX distribution has been presented in supplementary table 1, 2 and 3 for different time points. The vertical axis of figure 6, 7 and 8 vertical axis has been changed to “DOX concentration in Fluorescence Intensity (RFU/mg)”.

CYP is a prodrug which is converted to acrolein and phosphoramide mustard in presence of cytochrome P450 to exert the toxic effect, limiting the accuracy of drug measurement in the different organs after administration. That’s why we used doxorubicin as a model drug owing to its fluorescence properties for a comprehensive biodistribution analysis over a period of time.

The weight data of various tissues (liver, spleen, heart, lungs, brain, kidney and tumour) after treating mice with DOX and DOX-loaded NPs has been added in the supplementary table 4. The weights of different organs did not show any difference among the mice treated with only DOX and DOX-loaded NPs for 2 hr and 24 hr. The same time period was maintained to estimate the distribution of drug in the different organs. And the data of weight of various tissues (liver, spleen, heart, lungs, brain, kidney and tumour) after treating with CYP and CYP-loaded NPs has been added in Figure 2.

We did not add the data for AST, ALT, CK-MB markers as those were not approved to investigate by Monash Animal Ethics Committee. Additional biochemical markers including ALT were measured in the earlier study, with no significant differences observed among the CA-treated and untreated animals.

Yamamoto H, Wu X, Nakanishi H, Yamamoto Y, Uemura M1, Hata T, Nishimura J, Takemasa I1, Mizushima T, Sasaki J4, Imazato S4, Matsuura N5, Doki Y1, Mori M1. A glucose carbonate apatite complex exhibits in vitro and in vivo anti-tumour effects. Sci Rep. 2015 Jan 13;5:7742. doi: 10.1038/srep07742.

FE-SEM analysis of CYP and DOX – loaded CMCA NPs has been included.

The contents of Tables 6 and 7 have been explained in details (line 290-292).

The higher accumulation of DOX in cancer cells and the lower accumulation of DOX in heart (figures 7 and 8) have been elaborated in discussion section (line 383–391).

The drug free NPs were formulated separately, and the fluorescence intensity of only NPs was deducted from the DOX-loaded NPs fluorescence intensity during calculation to nullify the effect of the external environment.

The manuscript has been checked carefully to avoid all the typos.

Reviewer 2 Report

This manuscript present results in a model of breast cancer in Female Balb/c mice using drug carbonate apatite nanoparticles loaded with cyclophosphamide. The manuscript is well-written, easy to follow, lack of edition mistakes and both the title and the abstract clearly present the question posed by the authors. The experimental methodology is well described. In addition, the authors carried out a biochemistry analysis to assess sub-chronic toxicity. Importantly, taking advantage of the fluorescence properties of the doxorubicin as a model drug, a comprehensive biodistribution analysis over time is shown, including data about possible pathways of clearing.

However, I miss only few issues, but that they are key to understand the results showed.

Nanomaterials used needs a better characterization, because:

- lines 296-300: authors claim drug binding favoured by what I guess is an electrostatic attraction (anionic domains of the NP, cationic domain of the drug). Thus, surface charge (e.g. by means of Z-Potential measurement) should be added to support the explanation of the binding mechanism. The statement of the authors (line 260-262) that “FE-SEM characterization was performed to analyze the surface property, size and the binding”. FE-SEM does not characterize such surface property in a proper way.

- The SEM images show NPs rather polydisperse. The use of other techniques to provide more data about the overall size distribution of the NPs (e.g. photon correlation spectroscopy (Dynamic Light Scattering) would contribute to a better understanding of this characteristic of the materials employed.

Along with the previous comment, from 296 it reads “the size CA NPs, CMCA NPs and α-KAMCA NPs prepared by using salt precipitation technique resulted in smaller particle sizes which were, respectively, 428.4 ± 21.70 nm, 163 ± 10.96 nm and 291.5 298 ± 10.60 nm” i.e. the materials used have sizes ranging from “180 to 429”. To me, it is unclear why such different materials have similar binding affinities for the drug, (since they have been used at the same concentration (0.3 mg/kg BW)). Since binding is through the surface, at the same mass and different size, different surface areas are obtained...maybe because the polidispersity of the materials total surface area is similar, or different binding mechanism may apply?

As authors measure fluorescence of doxorubicin, the correlation of fluorescent intensity and NP biodistribution is not that straightforward. Perhaps doxorubicin has been released in serum, thus explaining the observed fluorescence in organs where NPs are not expected (e.g. the brain). I suggest to conduct some experiment on drug release (e.g. in model physiological media, and in acidic media since the author point out that (line 311) “The influence of pH-sensitive dissolution of CA, CMCA and α-KAMCA NPs ensured drug release at acidic microenvironment [28].” Or at least, expand the discussion with more details about potential doxorubicin release from the NPs before reaching the target.

Author Response

Answer:

The measurement of surface charge and particle size (through DLS system) for NPs and drug-loaded NPs have been published in our previous manuscripts.

Mehbuba Hossain, S.; Chowdhury, E.H. Citrate- and Succinate-Modified Carbonate Apatite Nanoparticles with Loaded Doxorubicin Exhibit Potent Anticancer Activity against Breast Cancer Cells. Pharmaceutics 2018, 10, doi:10.3390/pharmaceutics10010032.

Hossain, S.M.; Shetty, J.; Tha, K.K.; Chowdhury, E.H. alpha-Ketoglutaric Acid-Modified Carbonate Apatite Enhances Cellular Uptake and Cytotoxicity of a Raf- Kinase Inhibitor in Breast Cancer Cells through Inhibition of MAPK and PI-3 Kinase Pathways. Biomedicines 2019, 7, doi:10.3390/biomedicines7010004.

Hossain SM, Mozar FS, Chowdhury EH. Citrate association dramatically reduces diameter with concomitant increase in uptake of drug-loaded carbonate apatite particles. Journal of Nanoscience and Nanotechnology. 2018 April 26. [Ref no. - 2018-381] Accepted). In the discussion section we added the explanation based on surface charge, polydispersity index, pH sensitivity and DLS.

Round 2

Reviewer 1 Report

The authors respond carefully to my comments.  However, there are still several problems before publication in Cancers.

1) In figure 2C, the weight of the liver was greater in various DOX-treated groups than in healthy group. The authors should clearly explain why this phenomenon occurred.  Isn't it common for liver weight to decrease when DOX causes hepatic damage?

2) In supplementary table 4, authors should add the data of weight of various tissues (liver, spleen, heart, lungs, brain, kidney) in mice which were not treated with DOX.

3) I am very concerned about the slight environmental changes (e.g. pH et al.) in the solution of DOX that could change the fluorescence intensity of DOX itself due to physical properties of DOX.  The authors should clearly show the rationale for this not being possible.  The authors may show that the results of DOX concentration measured by the microplate reader are consistent with the results measured by the HPLC method.

Author Response

Answer:

1. Figure 2C represents the data for healthy mice, Cyclophosphamide (CYP) treated mice as a control and the mice treated with CYP-loaded different NPs as treatment groups. It was mentioned in the legend to the figure. However, we have now mentioned about CYP in the figure caption.

The weights of the liver and spleen were found greater in both untreated and drug-treated tumour-bearing mice than in healthy mice, which could be due to granulocytic hyperplasia (i, ii) in those organs of 4T1-induced breast tumour model.

  i. Exp Mol Pathol. 2007 Feb;82(1):12-24. Epub 2006 Aug 17. Murine mammary carcinoma 4T1 induces a leukemoid reaction with splenomegaly: association with tumor-derived growth factors. DuPre' SA, Hunter KW Jr.

  ii. Int J Exp Pathol. 2007 Oct;88(5):351-60. The mouse mammary carcinoma 4T1: characterization of the cellular landscape of primary tumours and metastatic tumour foci. DuPré SA, Redelman D, Hunter KW Jr.

2. In supplementary table 4, the data of weight of various tissues (liver, spleen, heart, lungs, brain, kidney) in mice which were not treated with DOX has been added.

3. For biodistribution analysis all the organs had been collected in lysis buffer of pH 7.4 (4.7 methodology section). The possible limitation and the rationality of using DOX for measuring drug concentration has been added in the discussion section. Secondly, doxorubicin fluorescence is only weakly pH-sensitive. Fluorescence intensity may change independently of concentration when doxorubicin dissolves into a medium of different dielectric constant, such as a non-aqueous environment [iii], which was not applicable to our case.

    iii. Swietach, P.; Hulikova, A.; Patiar, S.; Vaughan-Jones, R.D.; Harris, A.L. Importance of intracellular pH in determining the uptake and efficacy of the weakly basic chemotherapeutic drug, doxorubicin. PloS one 2012, 7, e35949, doi:10.1371/journal.pone.0035949.

Reviewer 2 Report

The authors have addressed all the comments in the revised manuscript.

I still think that a more exhaustive drug release study would complete the whole story presented in this work. However, in the way it is discussed is clear enough and a fine-tuning of the release studies can be a matter for a further work.

Author Response

Answer:

As we already have published drug release study from different NPs.

Now we have included in Figure 11  NPs dissolution study at different pHs (4.12 methodology section, 2.7 results section) with explanation in the discussion.

Round 3

Reviewer 1 Report

The authors respond carefully to my comments.  Therefore, I recommend that the article is published in the journal of "Cancer".